# Sexual Function After Laser Therapy for Penile Cancer: A Systematic Review

**DOI:** 10.3390/cancers17233737

**Published:** 2025-11-22

**Authors:** Pouya Ariamanesh, Mateusz Czajkowski, Marcin Matuszewski

**Affiliations:** 1Student Scientific Circle at the Department of Urology, Faculty of Medicine, Medical University of Gdańsk, Marii Skłodowskiej-Curie 3a, 80-210 Gdańsk, Poland; austin.ariamanesh@gumed.edu.pl; 2Department of Urology, Medical University of Gdańsk, Smoluchowskiego 17, 80-214 Gdańsk, Poland; marcin.matuszewski@gumed.edu.pl

**Keywords:** quality of life, patient satisfaction, sexual function, penile sparing, laser therapy, CO_2_ laser, Nd:YAG, Tm:YAG, penile cancer

## Abstract

Penile cancer is a rare malignancy that can significantly affect patients’ sexual function and psychological well-being. Increasingly, penile-sparing approaches are used to balance oncologic control with preservation of quality of life. Among these, laser therapy offers a minimally invasive option for early-stage disease, yet evidence on sexual outcomes remains fragmented. In this systematic review, we evaluated sexual function, patient satisfaction, and oncological outcomes after CO_2_, Nd:YAG, and Tm:YAG laser therapies for penile cancer. Across 11 studies involving 593 patients, erectile function was generally maintained in most patients, sexual satisfaction was generally acceptable, and cosmetic outcomes were largely favorable. However, sexual outcomes were inconsistently reported, often using non-validated tools, and data for advanced tumors were limited. Laser therapy is a valuable option for early-stage disease, but standardized outcome measures and long-term data are needed to fully define its functional benefits.

## 1. Introduction

Penile cancer, although rare, poses substantial physical and psychological challenges for patients. According to the 2020 data from Global Cancer Registries, there were around 36,000 newly diagnosed cases of penile cancer worldwide, along with nearly 13,000 reported fatalities [1]. Moreover, penile cancer constitutes less than 1% of all cancers among men in the United States, with an estimated 2100 new cases diagnosed annually [2]. However, its prevalence is notably higher in certain regions, including parts of Africa, Asia, and South America, where it can account for up to 10% of male cancers [3].

Penile cancer treatment modalities can be broadly classified into two categories: penile-sparing and non-penile-sparing approaches. Penile-sparing treatments are designed to conserve as much of the penile structure and function as possible utilizing techniques such as laser therapy; glansectomy; or partial penectomy with reconstruction using split-thickness skin grafts, Mohs micrographic surgery, and radiation therapy. In contrast, non-penile-sparing treatments involve more extensive surgical interventions, such as partial or total penectomy, which can significantly impact urinary and sexual function [4].

Historically, surgical management of penile cancer required a 2 cm free margin to ensure oncologic safety, often resulting in considerable functional and psychological morbidity. Subsequent evidence has demonstrated that margins greater than 1 mm are sufficient to achieve oncologic control while preserving penile anatomy and function [5]. This paradigm shift has led to the increasing adoption of penile-sparing surgery, balancing effective tumor eradication with improved postoperative quality of life, including sexual and urinary outcomes. Recent systematic reviews, such as that by Fallara et al. (2024), have reinforced the expanding role of penile-sparing approaches in localized disease, demonstrating favorable oncologic outcomes across more than 10,000 patients [6].

Despite this increasing interest in organ-sparing strategies for penile cancer, the current literature lacks a cohesive synthesis of how laser-based treatments impact long-term sexual function and patient-reported outcomes. Many existing studies prioritize recurrence and survival metrics while underreporting functional and psychosocial endpoints, such as erectile function, orgasm, and patient satisfaction.

Additionally, there is no established consensus on which laser modalities best balance oncologic control with quality-of-life preservation, especially in early-stage disease. Each laser modality interacts with biological tissue through distinct photothermal mechanisms, which may influence postoperative healing and functional outcomes. The CO_2_ laser emits infrared light at 10.6 μm, resulting in superficial vaporization with a penetration depth of approximately 0.05 mm, with a thermal coagulation zone of approximately 0.5 mm and minimal collateral thermal injury, allowing precise excision and rapid mucosal recovery [7]. This produces an extremely narrow zone of thermal damage, but the residual thermal energy is not enough to provide hemostasis in vascularized tissues.

In contrast, the Nd:YAG laser belongs to the group of solid-state lasers which operates at 1.064 μm with deeper penetration (4–6 mm) and greater coagulative necrosis, which, while advantageous for hemostasis and oncologic control, may increase the risk of damage to adjacent neurovascular structures relevant to erectile and sensory function [8,9].

The Tm:YAG laser (1.94 μm) exhibits intermediate absorption and shallow penetration (~0.3–0.5 mm), combining efficient cutting with excellent coagulation and limited thermal spread [9,10]. These biophysical differences may translate into variable effects on scarring, sensitivity, and erectile function after treatment. This gap complicates clinical decision-making, leaving patients with limited evidence to guide treatment decisions.

The aim of this systematic review was to evaluate the impact of laser-based penile-sparing surgeries (PSSs) on the sexual function of patients with penile cancer. Specifically, the objective was to assess how laser therapy affects key domains of sexual health, including erectile function, libido, penile length, and overall sexual satisfaction. Another aim was to identify factors associated with variability in sexual outcomes following treatment, such as laser types and whether their tissue-interaction profiles have clinically meaningful implications, treatment parameters, tumor stage, patient demographics, and follow-up duration. By synthesizing the available evidence, we intend to provide clinically relevant insights to guide urologists and oncologists in patient counselling, treatment planning, and shared decision-making, especially regarding the functional and psychosocial consequences of laser therapy. This systematic review was carried out in accordance with the Preferred Reporting Items for Systematic Reviews and Meta-Analyses (PRISMA) 2020 [11] guidelines to ensure a rigorous and transparent reporting process.

## 2. Materials and Methods

### 2.1. Eligibility Criteria

Studies were eligible for inclusion if they involved adult patients diagnosed with penile squamous cell carcinoma who underwent laser-based therapy as a primary treatment modality. Accepted laser modalities include carbon dioxide (CO_2_), neodymium-doped yttrium aluminum garnet (Nd:YAG), and thulium-doped yttrium aluminum garnet (Tm:YAG) and similar technologies used with curative or organ-sparing intent. To qualify, studies were required to report at least one relevant sexual function outcome, such as erectile function, depth of penetration, libido, orgasmic function, or overall sexual satisfaction. Eligible study designs included randomized controlled trials (RCTs), prospective or retrospective cohort studies, case–control studies, conference proceedings, and observational analyses. For the purpose of this review, while systematic reviews and guidelines offered invaluable insight, they were excluded due to lack of primary data. Instead, their references were manually screened for the primary sources of their data. Only articles published from January 2000 to 5 February 2025 were considered to ensure the inclusion of contemporary treatment practices and technologies. Furthermore, articles written in languages other than English were initially translated using automated tools (Google Translate; Google; Mountain View, CA, USA). If the preliminary assessment indicated potential, they would be sent for professional translation.

Studies were excluded if they were non-human experiments (e.g., animal or in vitro studies), focused exclusively on non-laser interventions (e.g., surgery, radiotherapy, or chemotherapy without any laser therapy component), or failed to report sexual function outcomes. Additionally, single case reports, narrative reviews, editorials, and expert opinions were excluded due to insufficient methodological rigor.

### 2.2. Outcome Measures

The included studies were categorized by the type of laser therapy used, allowing for subgroup comparisons between CO_2_, Nd:YAG, Tm:YAG, and other laser modalities. This structured classification supports a clearer interpretation of the impact of each treatment on sexual function and facilitates meaningful cross-study comparisons of the results. Outcomes such as patient satisfaction, sexual function, libido, cosmetic outcome, function retention, partner satisfaction, and oncological control were assessed for each laser type.

### 2.3. Search Strategy

A systematic search was performed in accordance with the PRISMA 2020 guidelines. We searched PubMed, Embase, Scopus, Google Scholar, ClinicalTrials.gov, and ScienceDirect using a combination of Medical Subject Headings (MeSH) and free-text terms pertaining to penile cancer, laser therapy, and sexual function. A further search was conducted on conference databases such ASCO, EAU, and AUA for abstracts. The search was limited to those matching the previously mentioned eligibility criteria. The full search strategies for each database are provided in Table 1. Reference lists of included studies were also screened manually.

### 2.4. Selection Process

Two independent reviewers screened the titles and abstracts for eligibility. Full-text articles of potentially relevant studies were assessed against the inclusion and exclusion criteria of this review. Discrepancies between the reviewers were resolved through discussion or consultation with a third reviewer if needed. No automation or artificial intelligence tools were used in study selection or data extraction; duplicates were removed manually.

### 2.5. Data Collection Process

Data were extracted independently by two reviewers using a standardized data extraction form in accordance with the key data items. Any discrepancies were resolved through consensus. If necessary, the authors of the original studies were contacted for missing information.

### 2.6. Data Items

Primary outcomes included erectile function, sexual satisfaction, libido, penetration, cosmetic outcome, function retention, oncological control, and patient satisfaction. When outcomes were not assessed with validated tools (e.g., the International Index of Erectile Function), they were extracted but considered low-certainty.Secondary variables included participant characteristics (age and stage of penile cancer), intervention details (type of laser, make, and mode), and study characteristics (year, design, sample size, and follow-up duration). Assumptions for missing data were clearly stated in the manuscripts.

Data regarding study design, patient characteristics, tumor stage, laser modality, and outcome measures were systematically extracted. Sexual and functional outcomes were categorized according to the assessment instruments used to enhance comparability across studies. These included (1) validated tools, such as the International Index of Erectile Function (IIEF), Life Satisfaction Checklist-11 (LiSat-11), Hospital Anxiety and Depression Scale (HADS) or the European Organization for Research and Treatment of Cancer Quality of Life Questionnaire-Core 30 (EORTC QLQ-C30) if available; (2) non-validated or author-derived instruments, including study-specific questionnaires and numerical rating scales without prior validation; and (3) qualitative assessments, encompassing structured or semi-structured interviews and patient self-reports describing sexual activity, satisfaction, or changes in erectile performance. This classification allowed consistent evaluation of methodological heterogeneity and facilitated interpretation of sexual function outcomes across different studies.

### 2.7. Risk of Bias and Quality Assessment, Effect Measures, and Data Synthesis

The risk of bias was assessed using the Newcastle–Ottawa Scale (NOS) for observational studies, covering three domains: selection (4 points), comparability (2 points), and outcome (3 points). Two independent reviewers conducted these assessments and resolved disagreements through discussion or consultation with a third reviewer. Certainty of evidence was appraised using Grading of Recommendations Assessment, Development, and Evaluation (GRADE). Because of heterogeneity in study designs and outcome reporting, no formal meta-analysis was performed. Therefore, we report crude proportions (n/N and %) only, without confidence intervals. We summarized crude proportions and undertook subgroup comparisons by laser type, tumor stage, and follow-up duration. Reporting bias was assessed qualitatively by reviewing the gray literature and trial registries.

## 3. Results

### 3.1. Study Selection

The search retrieved 192 records from databases and 16 from the gray literature. After de-duplication and screening, 83 full texts were assessed; 11 studies met the eligibility criteria and were included. The PRISMA flow diagram is shown in Figure 1. The most common reasons for exclusion were absence of laser therapy (surgery/radiotherapy only), focus on HPV or phototherapy, lack of objective sexual function outcomes, and guidance documents without primary data [12,13,14].

### 3.2. Study Characteristics

Across the 11 included studies, 593 patients with penile cancer who underwent laser therapy as part of their treatment were evaluated. Most patients presented with early-stage disease: carcinoma in situ (Tis) was reported in 202 patients (33.9%) and T1 stage in 286 patients (48.2%). T2 tumors were identified in 78 patients (13.2%), whereas T3 disease was rare and was found in only 2 patients (0.3%). The tumor stage was not clearly specified in 25 cases (4.2%). Overall, the data reflect the predominance of superficial, organ-confined lesions suitable for penile-sparing laser therapy. The sample sizes ranged from 8 to 224 patients, with most studies involving 20–60 participants. The mean and median ages of the patients ranged from 37 to 73 years.

Among the included studies, all except one were retrospective in design, reflecting the rarity of penile cancer and the predominance of single-center experiences. Specifically, eight studies were retrospective (including case series, cohort, and interview-based analyses), one was a prospective observational study, one was a retrospective cohort study, and one was a retrospective review. No randomized controlled trials (RCTs) were identified. This distribution underscores the descriptive and exploratory nature of the current evidence base for laser-based penile-sparing therapy. The study characteristics, study design, and main findings are summarized in Table 2.

Two studies used CO_2_ laser therapy, focusing on early-stage penile cancer, particularly erythroplasia of Queyrat and squamous cell carcinoma (SCC) [15,16]. Three other studies utilized Nd:YAG, addressing both invasive and non-invasive penile squamous cell carcinoma [17,18,19]. Predictably, only one study employed Tm:YAG laser therapy, focusing primarily on early-stage penile cancer [20]. Lastly, five remaining studies assessed the use of combination of lasers or unspecified laser types [21,22,23,24,25]. A summary of the extracted data from CO_2_ laser, Nd:YAG laser, Tm:YAG laser, and other types of laser treatments is provided in the Appendix A, respectively.

Follow-up durations varied substantially across studies and were closely linked to the type of laser used. CO_2_ laser studies reported both short- and long-term outcomes, ranging from 66 months [15] to 10 years [16], with others reporting medians of 4.5 to 5.5 years. Nd:YAG laser studies had the longest follow-ups overall, including a mean of 87 months [17] and medians up to 70 and 46 months [18,19]. Potassium titanyl phosphate laser (KTP) [24] and Tm:YAG [20] studies had shorter median follow-ups of 28 and 24 months, respectively. Overall, while some studies provided long-term outcome data, others offered only short-to-mid-term follow-up, contributing to heterogeneity in the evidence base.

#### Risk of Bias in Studies

On average, the included studies demonstrated a moderate risk of bias. Two studies were rated as low-risk [22,23,24], while four studies were rated as low-to-moderate-risk [15,21,23,24]. The remaining studies were assessed as moderate-risk, most often due to retrospective single-arm designs, a lack of comparator groups, and the absence of statistical adjustment for confounders [16,17,18,19,20]. Common strengths included well-defined eligibility criteria and secure exposure ascertainment, whereas common limitations included limited follow-up in some series (<36 months) and no multivariable control for prognostic factors. A detailed breakdown of NOS scoring is presented in the risk of bias assessment in Table 3.

### 3.3. Results of Individual Studies

#### 3.3.1. Heterogeneity

Substantial heterogeneity was present across the included studies in terms of design, sample size, tumor staging, follow-up duration, laser modality, and outcome assessment. Tumor stages varied even within the same laser category, with some including only early-stage lesions (Tis/T1), while others incorporated more advanced cases (e.g., T2/T3). Additionally, the use of validated tools to assess functional, cosmetic, or psychological outcomes was inconsistent or absent in most studies. These variations introduce both clinical and methodological heterogeneity, limiting the reliability of cross-study comparisons and pooled interpretations.

The outcome instruments used across studies were highly heterogeneous, reflecting substantial methodological variability in how sexual function was defined and assessed. Only a minority of studies employed validated questionnaires, such as the IIEF, LiSat-11, or HADS, to quantify erectile or sexual outcomes objectively. The majority relied on non-validated, author-developed questionnaires or qualitative patient interviews, often lacking standardized scoring systems or reference thresholds. In several retrospective series, sexual function was inferred from patient self-reports or non-specific terms, which limits comparability and interpretability across studies. These methodological inconsistencies reduce the certainty of pooled evidence and underscore the need for standardized, validated outcome measures in future research. A detailed summary of the assessment instruments used in each study is presented in Table 4.

The individual study results are summarized in the results of individual syntheses in Table 5, which presents the key findings from each study, including the summary statistics for each group, effect estimates, and the precision of the estimates.

#### 3.3.2. Erectile Function Outcomes by Laser Type

Erectile function preservation ranged from 59% to 100% across studies. Among Nd:YAG laser cohorts, preservation was 59.3–71.9% [17,18], and about 67% when recalculated from Frimberger [19]). CO_2_ laser monotherapy showed full preservation, with no erectile dysfunction reported [15,16]. In combined CO_2_ + Nd:YAG series, rates were variable: Skeppner (2008) reported generally reduced function [21], while Windahl reported 72% unaltered, 22% decreased, and 6% improved outcomes [23]. Tm:YAG laser maintained function unaltered post-treatment [20]. Overall, erectile function was preserved in about 68% of patients across all laser modalities.

#### 3.3.3. Sexual Function

Sexual function preservation was generally high across all laser modalities, though outcome definitions and measurement tools varied. Nd:YAG laser studies reported pooled crude preservation rates of 64.3%, with individual study estimates ranging from 59.3% to 75.0%, and some studies noting improvements in penile preservation and minimal impact on orgasmic function [17,18,19]. Combined CO_2_ therapies reported 50–72% satisfaction, with most patients resuming sexual activity within weeks to months. However, one study found aspects of sexual life, such as manual stimulation/caressing and fellatio, decreased markedly after treatment. Moreover, it reported that 10/21 (47.6%) sexually active men had dyspareunia before treatment, which was reduced to 2/17 (11.7%) at follow-up. In contrast, 4/21 (19.0%) patients could not achieve penetrative sex after the treatment but resorted to other forms of sexual activity [21,22,23]. Another study reported a significant decrease in sexual satisfaction after treatment (*p* = 0.039) [22]. Tm:YAG laser therapy showed promising outcomes with 82% of patients maintaining intercourse and high sexual function scores at one-year follow-up [20]. Overall, all penile-sparing studies reported better sexual satisfaction than non-penile-sparing techniques. Although sexual function is generally well-preserved, reported outcomes vary, likely reflecting the multifactorial influences such as patient characteristics, underlying causes, and demographic differences. A summary of sexual function preservation by laser type is provided in Appendix A.

#### 3.3.4. Cosmetic Outcomes and Body Image Issues

Cosmetic outcomes were consistently favorable across all laser modalities, with minimal scarring and high satisfaction. In Nd:YAG series, cosmetic satisfaction ranged from 80% to 100%, and several studies noted improved body image and psychological outcomes compared with penectomy [17,18,19]. Schlenker et al. reported “excellent” cosmetic results, while Frimberger and Tewari found all patients satisfied with appearance and penile preservation. CO_2_ monotherapy achieved uniformly positive results; Bandieramonte and Conejo-Mir both described “excellent” or “high” satisfaction, no scarring, and full preservation of penile form [15,16]. With Tm:YAG therapy, Musi observed 78.2% reporting no change in penile length and over 80% cosmetic satisfaction [20]. Combined CO_2_ + Nd:YAG studies reported similarly high outcomes; Windahl found 78% of patients satisfied or very satisfied, with no deformity, and Skeppner reported body image issues and life-satisfaction scores comparable to the general population [21,22,23]. Overall, laser-based penile-sparing therapy achieves excellent cosmetic preservation and improved body image relative to non-penile-sparing approaches (Appendix A) [16,17,18,19,20].

#### 3.3.5. Extent of Tissue Preservation

While the extent of penile tissue preservation is primarily determined by the tumor’s clinical stage (cT) and corresponding surgical indication, the uniformly high preservation rates observed across laser-treated cohorts (>96%) suggest that laser techniques allow for precise ablation with minimal collateral thermal injury. In CO_2_ laser studies, organ preservation ranged from 96% to 100% [15,16], while Nd:YAG and Tm:YAG series consistently reported 100% preservation in appropriately selected early-stage patients [17,18,19]. Comparable results were also achieved with combined CO_2_ + Nd:YAG therapy [21,22,23]. While such high preservation rates were anticipated given that most included patients had carcinoma in situ (CIS), Tis, or T1 disease, the near-complete preservation observed suggests that the lasers caused no appreciable additional thermal damage beyond the tissue margins required for oncologic clearance. Collectively, these findings indicate that, although tumor stage dictates the extent of resection, the laser’s precision and limited depth of thermal injury contribute to excellent immediate tissue preservation and favorable cosmetic outcomes in early-stage penile cancer. Detailed data on penile preservation outcomes are presented in Appendix A.

#### 3.3.6. Clinical Outcomes

Oncologic control following laser-based penile-sparing therapy was favorable across all modalities. Recurrence rates varied across laser types, influenced by differences in tumor staging and follow-up duration. CO_2_ laser studies reported relatively consistent outcomes: 12.5% in Conejo-Mir et al. and 17.4% in Bandieramonte et al. [15,16]. Nd:YAG studies showed greater variability, with one reporting a high recurrence rate of 42.6%, likely influenced by longer follow-up and inclusion of higher-stage tumors [17], while others [18,19] reported substantially lower rates of 6.25% and 6.9%, respectively. The Tm:YAG study reported a recurrence rate of 17.4%, including 13.0% invasive recurrences, com-parable to CO_2_ laser outcomes but based on a shorter follow-up period [20]. Five-year survival outcomes following laser-based therapies were favorable across modalities. Five-year survival was reported 100% in one study for CO_2_ laser therapy [16], Across studies employing Nd:YAG laser therapy for early-stage penile carcinoma, 5-year disease-specific survival ranged from 98% to 100%. Refs. [17,18,19], and 95.6% for combined CO_2_ + Nd:YAG therapy, with disease-specific survival ≥ 95% where reported [21,22,23]. Tm:YAG therapy achieved 100% survival in early-stage disease at 2 years [20]. Overall, all laser types demonstrate acceptable oncologic control in early-stage disease but highlight the importance of rigorous, long-term follow-up to detect delayed recurrences and validate long-term efficacy. Details regarding the recurrence and the clinical outcomes are given in Appendix A [15,16,17,18,19,20,21,22,23].

#### 3.3.7. Postoperative Pain and Complications

Nd:YAG laser studies reported fast healing and minimal complications [17,18,19]. Tewari observed no need for parenteral analgesia [18], and both Tewari and Frimberger reported no major postoperative issues such as infections or persistent pain [18,19]. Schlenker noted fast recovery, although follow-up details on short-term healing were limited [17]. CO_2_ laser studies demonstrated consistent short-term recovery benefits [15,16]. Full epithelialization was observed within 14–28 days with minimal discomfort [16], and pain was resolved within three days [15]. No major complications were noted across these studies. Tm:YAG laser studies described mild discomfort localized to the external meatus, with healing typically completed by secondary intention within five weeks. Postoperative edema and transient dysuria were minimal and self-limited [20]. Combined CO_2_ and Nd:YAG laser therapy resulted in longer recovery, with full healing typically requiring up to three months, and dyspareunia was reported in 10% of cases [23]. However, one study noted a reduction in dyspareunia following treatment [25,26]. Overall, across laser modalities, postoperative recovery was characterized by minimal pain, rapid healing, and a low incidence of complications.

### 3.4. Reporting Biases

Most included studies were retrospective and non-randomized, which increases the risk of selective reporting and incomplete outcome data. Common issues were omission of standardized sexual function instruments, inconsistent follow-up durations, and missing data for recurrence and survival endpoints. A detailed summary is provided in Table 6.

#### 3.4.1. CO_2_ Laser Therapy Studies

Sexual outcomes were frequently described qualitatively with no validated instruments (e.g., IIEF-5). Reported recurrence ranged from ~12.5 to 17.4%, but interpretability was limited by small samples (e.g., *n* = 8) and variable follow-up. Overall, reporting bias was moderate–high [15,16].

#### 3.4.2. Nd:YAG Laser Therapy Studies

Recurrence estimates varied widely (~6–43%), likely reflecting differences in stage mix and follow-up length; none used validated measures for erectile function, libido, or orgasm [17,18,19]. Reporting bias was high due to inconsistent methodology and outcome definitions.

#### 3.4.3. Tm:YAG Laser Therapy Study

Only one study was available; recurrence was 17.4% (4/23 evaluable) at short follow-up (median 24 months), without standardized sexual function tools or a comparator. Reporting bias was very high [20].

#### 3.4.4. Other Studies

Sexual function outcomes were inconsistently reported and lacked validated metrics. For example, one reported 72% erectile function adequacy but used no validated scales and noted a drop in satisfaction from 61% to 41% post-treatment but provided no comparative data [21]. Another study suggested improved satisfaction with less invasive therapies but offered no structured analysis. Reported recurrence rates ranged from 19% to 25%, with follow-up periods spanning 12 months to 15 years. These methodological limitations contribute to a risk of reporting bias, primarily due to lack of standardization, incomplete long-term data, and potential underreporting of adverse events [21,22,23,24,25].

### 3.5. Certainty of Evidence

Overall, the certainty of evidence across outcomes was low to moderate, reflecting the predominance of retrospective, non-randomized studies with methodological limitations. Certainty of evidence for each primary and secondary outcome was assessed using the Grading of Recommendations, Assessment, Development, and Evaluation (GRADE) framework in accordance with Cochrane and GRADE Working Group guidance. Evidence derived from observational studies was initially rated as low certainty and subsequently downgraded or upgraded based on five domains: risk of bias, inconsistency, indirectness, imprecision, and publication bias. Upgrading was considered when consistent large effects or dose–response relationships were observed across multiple studies. Final certainty were determined for each outcome, including erectile function, sexual satisfaction, penile preservation, recurrence, and survival. The detailed assessment of certainty of evidence for each outcome is provided in Table 7.

The evidence for sexual function preservation after laser therapy for penile cancer is of low certainty due to methodological issues, including the lack of standardized assessments and reliance on subjective patient-reported outcomes. Reporting is inconsistent—some studies note general satisfaction (e.g., 50% in Windahl et al.) [23], while others report only on resumed activity (e.g., 82.6% in Musi et al.) [20]. Most lack comparative groups, limiting conclusions. Although laser therapy may preserve function better than surgery, more rigorous, standardized studies are needed.

Penile tissue was successfully preserved with moderate certainty as multiple studies consistently reported high rates (96–100%) without major inter-study discrepancies, in line with expectations early-stage cancer and with organ-sparing techniques [20].

Certainty for oncologic control, which includes overall survival and disease-free survival, is moderate. This assessment is based on several factors: amongst the reported articles, 5-year overall survival rates were generally high, ranging between 80% and 95%. Certainty for recurrence rates was low, with estimates ranging from 6% to 42% and substantial variability in follow-up duration (under 2 years to >10 years) and tumor staging inclusion. Furthermore, disease-free survival was reported inconsistently, making it difficult to evaluate long-term effectiveness.

Patient satisfaction and psychosocial outcomes were rated very low due to the absence of validated quality-of-life or partner satisfaction measures, with only occasional qualitative reporting [22].

## 4. Discussion

### 4.1. General Interpretation

This review supports laser therapy as an effective penile-sparing option for patients with early-stage penile cancer, demonstrating high rates of functional preservation. Across included studies, erectile and sexual function were maintained in approximately 60–100% of patients, and penile tissue was successfully preserved or undamaged in 96–100% of cases, with most reports also describing favorable cosmetic results and minimal morbidity. These outcomes indicate that laser therapy provides reliable functional preservation while maintaining oncologic safety, aligning with the growing clinical emphasis on quality of life in penile cancer management.

Compared to non-penile-sparing techniques such as partial penectomy and glansectomy without reconstruction, laser therapy consistently results in superior sexual and psychological outcomes. Moreover, while traditional surgeries may achieve superior oncologic control, they often entail substantial physical and emotional consequences, including loss of erectile function, altered body image, and diminished sexual activity [27]. Laser approaches, particularly CO_2_ and Nd:YAG, preserve key anatomical structures and are thus associated with better maintenance of penile sensation, appearance, and sexual function.

Cosmetic and psychosocial outcomes are particularly favorable with laser therapy. High satisfaction with genital appearance, minimal scarring, and low distress levels were consistently observed across modalities. In contrast, patients undergoing partial or total penectomy frequently report dissatisfaction with genital aesthetics, and many experience long-term psychological effects that impair their intimacy and self-image [28]. However, it is important to acknowledge that glansectomy and penectomy are typically indicated for more advanced stages of penile cancer.

From an oncological perspective, laser-based penile-sparing therapy demonstrates recurrence and survival outcomes that are broadly comparable to other conservative and radical approaches when applied in appropriately selected patients. Across the included studies, local recurrence rates ranged from 6–19% [6]. Most recurrences were superficial and successfully managed with repeat laser ablation, thereby preserving penile anatomy and avoiding the morbidity of salvage partial or total penectomy. Reported five-year overall survival exceeded 90% across. Although recurrence rates may be somewhat higher than those seen with more radical procedures, the availability of repeat laser treatment and preservation of tissue integrity allow for flexible, staged management.

However, these favorable results are contingent upon rigorous and sustained postoperative surveillance, which is essential for detecting and managing local recurrence at an early stage. Limited access to specialist follow-up care, resource constraints, or patient non-adherence to long-term monitoring may undermine these outcomes. Moreover, patients who prioritize definitive reassurance over functional cost, may opt for more radical options.

Collectively, the evidence suggests that laser therapy achieves a favorable balance between oncologic control and functional preservation in early-stage penile cancer, but its long-term success depends on meticulous patient selection, adherence to structured follow-up, and shared decision-making within a multidisciplinary care framework [29].

Radiation-based therapies, including brachytherapy and external beam radiotherapy, offer organ preservation but are associated with longer treatment durations, higher rates of late toxicity (e.g., fibrosis, strictures), and inferior cosmetic outcomes in some cohorts [29,30]. In contrast, laser therapy is typically performed as a single or limited-session procedure with rapid healing, low complication rates, and the advantage of re-treatability without cumulative tissue damage.

Postoperative recovery was uniformly favorable, with most patients experiencing minimal pain, rapid epithelialization, and no significant complications. Healing times of 2 to 6 weeks were common, with only minor discomfort reported. These outcomes highlight the minimally invasive nature of laser ablation, especially when contrasted with the more prolonged recovery and wound care required after glansectomy or partial penectomy [27].

The comparative findings across laser modalities suggest that their distinct tissue-interaction profiles may have modest but clinically plausible implications for postoperative functional outcomes. The CO_2_ laser, characterized by shallow penetration (≈0.05 mm) and a narrow thermal coagulation zone, achieved the highest rates of erectile function preservation and the most rapid recovery of sexual activity [7,15,16]. In contrast, the Nd:YAG laser, with deeper optical penetration (4–6 mm) and greater coagulative necrosis, demonstrated slightly lower sexual satisfaction rates (59–72%) [17,18,19], findings consistent with its broader thermal spread and potential to affect subepithelial neurovascular structures [8,9]. The Tm:YAG laser, with intermediate absorption depth (0.3–0.5 mm), produced outcomes between those of CO_2_ and Nd:YAG with 82% of patients returning to sexual intercourse, aligning with its balanced cutting–coagulation properties [9,20]. Although these differences correspond to known biophysical mechanisms, current evidence remains suggestive rather than definitive, given the heterogeneity of study designs and the absence of standardized sexual-function assessment tools. Larger comparative studies are needed to determine whether these optical distinctions translate into durable clinical advantages.

From a clinical standpoint, laser-based PSTs are effective first-line treatments for early-stage penile cancer (Tis, T1), particularly for patients prioritizing functional outcomes. Their benefits include preserved penile structure, reduced psychological impact, and maintained sexual function. However, in patients with T2 or higher disease, recurrence risks are greater, and close monitoring is essential. These therapies should be presented during shared decision-making, highlighting the trade-offs between functional preservation and oncologic risk. Expanding access to laser therapy may also reduce delays in care caused by fear of radical surgery. Given the physical, psychological, and sexual impact of penile cancer, multidisciplinary care is essential. Urologists, oncologists, dermatologists, radiation oncologists, sexual health specialists, psychologists, and primary care providers should collaborate to deliver comprehensive care and long-term support. Integrating laser therapy into a team-based model enhances both oncologic outcomes and quality of life.

However, several important limitations were identified in the current evidence base on laser-based penile-sparing therapies. First, the majority of studies were retrospective, introducing selection bias and limiting control for confounding variables. Incomplete datasets and inconsistent outcome reporting further reduce confidence in the findings. Publication and reporting biases may have inflated positive results as adverse outcomes and treatment failures were rarely documented. Moreover, the lack of comparative groups in most studies limits the contextual interpretation of functional and oncologic outcomes. Second, there was substantial heterogeneity across studies regarding the tumor stage, laser modality, treatment parameters, and follow-up duration. This clinical and methodological variability complicates cross-study comparisons and precludes robust data synthesis. Third, sexual outcomes were typically assessed using non-validated, subjective measures, which limits comparability and clinical applicability. Follow-up protocols were also highly variable; while some studies incorporated imaging or biopsy, others relied solely on infrequent physical examinations. Adherence to surveillance protocols was rarely evaluated, despite its critical role in early recurrence detection. Finally, long-term oncologic data were limited. Although short-term control was acceptable, data beyond five years were sparse, and definitions of recurrence varied considerably between studies.

There is a clear need for randomized controlled trials comparing laser therapy to other PSTs (e.g., glansectomy, Mohs surgery) and to radical procedures. Long-term prospective cohort studies should evaluate recurrence, survival, sexual function, and psychological outcomes, with follow-up extending beyond 5–10 years to assess oncologic durability.

### 4.2. Conclusions

Laser therapy represents a functionally superior, low-morbidity alternative to conventional non-penile-sparing surgical approaches, including partial and total penectomy, for the management of early-stage penile cancer. In patients prioritizing the preservation of sexual function and aesthetic appearance, laser-based treatment achieves favorable functional and psychosocial outcomes while maintaining acceptable oncologic control. Although vigilant long-term surveillance remains essential, particularly in higher-stage disease, the overall balance between organ preservation, quality of life, and oncologic safety supports the use of laser therapy in appropriately-selected cases.

## 5. Other Information

This systematic review was conducted in accordance with the PRISMA 2020 guidelines [11] to ensure methodological rigor and transparency. The study protocol was retrospectively registered on the Open Science Framework (OSF), under a CC BY 4.0 license to enhance reproducibility and public accessibility (https://doi.org/10.17605/osf.io/gjaz6, accessed on 5 October 2024). OSF registration was selected as an appropriate alternative to PROSPERO, as the review focuses on functional and quality-of-life outcomes rather than interventional therapies, which fall outside PROSPERO’s eligibility criteria.

## Figures and Tables

**Figure 1 cancers-17-03737-f001:**
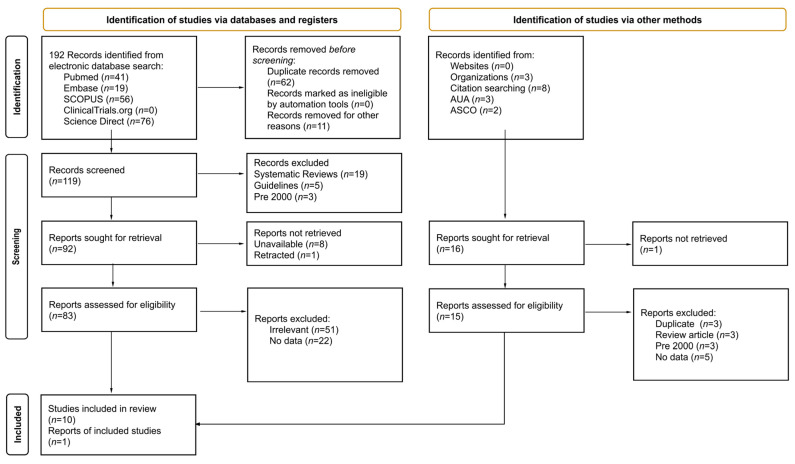
PRISMA diagram: Flowchart illustrating the study selection process for the systematic review, including the number of records identified, screened, assessed for eligibility, and included in the final analysis, along with the reasons for exclusion at each stage.

**Table 1 cancers-17-03737-t001:** Detailed search strings and Boolean combinations employed in each database are presented.

	Penile Cancer	(AND) Laser	AND Sexual Function	NOT
PubMed	((penile cancer) OR (penile carcinoma) OR (penile neoplasm) OR (penile squamous cell carcinoma) OR (penile SCC) OR (penile tumor) OR (penile malignancy) OR (penile lesion) OR (malignant penile neoplasm) OR (penile intraepithelial neoplasia) OR (PeIN) OR (penile squamous carcinoma) OR (carcinoma of the penis))	((laser therapy) OR (laser ablation) OR (CO_2_ laser) OR (Nd:YAG laser) OR (diode laser) OR (Holmium laser) OR (Thulium laser))	((sexual function) OR (erectile dysfunction) OR (sexual health) OR (sexual outcomes) OR (libido) OR (orgasm function) OR (penile sensitivity) OR (penetrative ability) OR (sexual quality of life) OR (SQoL) OR (quality of life) OR (sexual recovery) OR (post-treatment sexual function))	((prostate) OR (prostatic) OR (prostate cancer) OR (prostatic neoplasm) OR (prostate carcinoma) OR (BPH) OR (benign prostatic hyperplasia) OR (pearly))
Embase	(‘penile cancer’ OR ‘penile carcinoma’ OR ‘penile neoplasm’ OR ‘penile squamous cell carcinoma’ OR ‘penile SCC’ OR ‘penile tumor’ OR ‘penile malignancy’ OR ‘penile lesion’ OR ‘malignant penile neoplasm’ OR ‘penile intraepithelial neoplasia’ OR ‘PeIN’ OR ‘penile squamous carcinoma’ OR ‘carcinoma of the penis’)	(‘laser therapy’ OR ‘laser ablation’ OR ‘CO_2_ laser’ OR ‘Nd:YAG laser’ OR ‘diode laser’ OR ‘Holmium laser’ OR ‘Thulium laser’)	(‘sexual function’ OR ‘erectile dysfunction’ OR ‘sexual health’ OR ‘sexual outcomes’ OR ‘libido’ OR ‘orgasmic function’ OR ‘penile sensitivity’ OR ‘penetrative ability’ OR ‘sexual quality of life’ OR ‘SQoL’ OR ‘quality of life’ OR ‘sexual recovery’ OR ‘post-treatment sexual function’)	(‘prostate’ OR ‘prostatic’ OR ‘prostate cancer’ OR ‘prostatic neoplasm’ OR ‘prostate carcinoma’ OR ‘BPH’ OR ‘benign prostatic hyperplasia’ OR ‘pearly’)
Scopus	(TITLE-ABS-KEY(“penile cancer” OR “penile carcinoma” OR “penile neoplasm” OR “penile squamous cell carcinoma” OR “penile SCC” OR “penile tumor” OR “penile malignancy” OR “penile lesion” OR “malignant penile neoplasm” OR “penile intraepithelial neoplasia” OR “PeIN” OR “penile squamous carcinoma” OR “carcinoma of the penis”))	(TITLE-ABS-KEY(“laser therapy” OR “laser ablation” OR “CO_2_ laser” OR “Nd:YAG laser” OR “diode laser” OR “Holmium laser” OR “Thulium laser”))	(TITLE-ABS-KEY(“sexual function” OR “erectile dysfunction” OR “sexual health” OR “sexual outcomes” OR “libido” OR “orgasmic function” OR “penile sensitivity” OR “penetrative ability” OR “sexual quality of life” OR “SQoL” OR “quality of life” OR “sexual recovery” OR “post-treatment sexual function”))	(TITLE-ABS-KEY(“prostate” OR “prostatic” OR “prostate cancer” OR “prostatic neoplasm” OR “prostate carcinoma” OR “BPH” OR “benign prostatic hyperplasia” OR “pearly”))
ScienceDirect	(“penile cancer” OR “penile carcinoma”)	(“laser” OR “laser Therapy”)	(“sexual function” OR “satisfaction” OR “quality of life”)	(“prostate” OR “penectomy”)
ClinicalTrial.gov, ASCO, EAU, AUA	Penile cancer	Laser	Sexual Function	N/A
Google Scholar	Penile cancer	Laser	Sexual Function	Prostate

**Table 2 cancers-17-03737-t002:** Study characteristics and main findings. Details such as study design, sample size, laser type used, patient age, cancer stage, and key clinical outcomes across the included studies are included.

Author	Study Design	Sample Size	Laser Type	Age (Mean/Median)	Cancer Stage	Main Findings
Bandieramonte (2008) [15]	Retrospective Study	224	CO_2_	57 (range 20–83)	Tis (47.3%), T1 (52.7%)	CO_2_ laser excision under peniscopic control provides excellent oncological and functional outcomes for early-stage penile carcinoma
Conejo-Mir (2005) [16]	Retrospective Study	8	CO_2_	64 (range: 44–70)	Tis (100%)	CO_2_ laser therapy is an effective, safe, and cosmetically favorable treatment for erythroplasia of Queyrat
Schlenker (2010) [17]	Retrospective Case Series	54	Nd:YAG	57.6 (range: 25–89)	Tis (20.3%)), T1 (72.2%), T2 (7.4%)	Laser therapy offers organ preservation with functional benefits but carries a high recurrence rate
Tewari (2007) [18]	Retrospective Study	32	Nd:YAG	47 (range 32–67)	T1 (78.1%), T2 (21.9%)	Nd:YAG laser treatment for early-stage penile cancer provides good oncological control while preserving function
Frimberger (2002) [19]	Retrospective Study	29	Nd:YAG	55.1 (range: 30–93)	Tis (58.6%), T1 (34.5%), T2 (6.8%)	Nd:YAG laser therapy is an effective alternative to amputation for early-stage penile carcinoma. Excellent cosmetic and functional results
Musi (2018) [20]	Retrospective Study	26	Tm:YAG	61 (range: 54–72)	Tis (42.3%), T1 (30.7%), T2 (11.0%), T3 (0.7%)	Tm:YAG laser therapy preserves penile structure and function while providing effective oncological control. Good functional outcome, with a minor impact on patient’s quality of life
Skeppner (2008) [21]	Retrospective Interview Study	46	Combined CO_2_, Nd:YAG	63.5 (range: 34–90)	Tis (28.3%), T1 (30.4%), T2 (41.3%)	Laser treatment preserves function and quality of life; patients maintain sexual activity and life satisfaction
Skeppner (2015) [22]	Prospective Observational Study	29	Combined CO_2_, Nd:YAG	60 (range: 37–73)	T1 (41.4%), T2 (48.3%)	Life satisfaction matched the general population, but health and sexual satisfaction declined
Windahl (2004) [23]	Retrospective Study	67	Combined CO_2_, Nd:YAG	64 (range: 34–90)	Tis (31.3%), T1 (37.3%), T2 (31.3%),	Laser therapy provides high rates of functional and aesthetic satisfaction while preserving sexual function
Kokorovic (2021) [24]	Retrospective Cohort Study	58	KTP/CO_2_ or OSS + KTP	61.5 (range: 53.1–70.2)	Tis (25.8%), T1a (60.3%), T2 (13.8%)	Laser treatment allows organ preservation but can lead to reduced sexual function and satisfaction
Shaker (2023) [25]	Retrospective Review	20	Combined CO_2_, Nd:YAG	N/A	T1, T2	Laser ablation offers strong local control with better quality of life compared to more invasive procedures

**Table 3 cancers-17-03737-t003:** Risk of bias assessment. The methodological quality of the included studies using tools such as the Newcastle–Ottawa Scale (NOS) is summarized, with commentary on potential biases and limitations.

Study	Representativeness of Exposed Cohort	Selection of Non -Exposed Cohort	Ascertainment of Exposure	Outcome not Present at Start	Comparability (Design/Analysis)	Assessment of Outcome	Follow-Up Long Enough	Adequacy of Follow-Up	Total Stars	Risk of Bias	Comments
Bandieramonte 2008 [15]	+	−	+	+	−	+	+	+	7	Low-Moderate	Large consecutive series of early-stage penile cancer; clear CO_2_ laser protocol, long follow-up, minimal attrition; lacks comparator or confounder adjustment.
Conejo-Mir 2005 [16]	+	−	+	+	−	+	−	+	6	Moderate	Small retrospective case series of 8 EQ patients; standardized CO_2_ laser protocol; complete follow-up reporting but no comparator and unclear follow-up duration for all cases.
Schlenker 2010 [17]	+	−	+	+	−	+	+	+	6	Moderate	Retrospective single-center series; no comparator group and no confounder adjustment but strong long-term follow-up and outcome assessment.
Tewari 2007 [18]	+	−	+	+	−	+	+	+	6	Moderate	Retrospective single-center cohort; no comparator group or confounder adjustment but adequate follow-up and standardized exposure.
Frimberger 2002 [19]	+	−	+	+	−	+	+	+	6	Moderate	Retrospective single-center cohort; no comparator group or confounder adjustment but standardized laser protocol and long-term follow-up.
Musi 2018 [20]	+	−	+	+	−	+	−	+	6	Moderate	Retrospective consecutive case series; standardized Tm:YAG protocol; complete follow-up but relatively short median follow-up (24 months) and no comparator group or confounder adjustment.
Skeppner 2008 [21]	+	−	+	+	−	+	+	+	7	Low-Moderate	Consecutive localized penile cancer cases treated with laser; clear exposure documentation, adequate follow-up, minimal attrition; no comparator or confounder adjustment.
Skeppner 2015 [22]	+	+	+	+	++	+	+	+	9	Low	Prospective cohort with partner comparator group; controlled for major confounders; complete follow-up; low risk of bias.
Windahl 2004 [23]	+	−	+	+	−	+	+	+	7	Low-Moderate	Consecutive localized penile cancer cases treated with combined CO_2_ and Nd:YAG laser therapy; validated outcome measures; adequate follow-up; no comparator or confounder adjustment.
Kokorovic 2021 [24]	+	−	+	+	−	+	+	+	7	Low-Moderate	Large single-center OSS series with standardized protocols and histological confirmation; long-term follow-up subset; lacks comparator and multivariable confounder adjustment.
Shaker 2023 [25]	+	+	+	+	−	+	+	+	8	Low	Retrospective comparative cohort with three penile-sparing approaches; strong follow-up and complete data capture; no confounder adjustment.

‘+’ denotes that the study received a point for the respective criterion. ‘++’ denotes that the study received over one point for the respective criterion. ‘−’ denotes that the study did not receive a point for the respective criterion.

**Table 4 cancers-17-03737-t004:** Outcome measurement instruments used across included studies for assessment of sexual and functional parameters.

Author (Year)	Outcome Tool(s)	Instrument Type	Domains Evaluated
Bandieramonte et al. (2008) [15]	Clinical and peniscopic evaluation; patient interviews	Non-validated/Qualitative	Erectile function, sexual activity, cosmetic satisfaction
Conejo-Mir et al. (2005) [16]	Clinical observation; patient interview	Non-validated/Qualitative	Sexual and urinary function, cosmetic outcome
Schlenker et al. (2010) [17]	Structured follow-up interview	Non-validated/Qualitative	Erectile function, sexual activity, glans sensitivity
Tewari et al. (2007 [18]	Postoperative patient questionnaire	Non-validated/Qualitative	Erectile function, penetration ability, urinary and cosmetic outcomes
Frimberger et al. (2002) [19]	Physician-conducted interviews	Non-validated/Qualitative	Sexual activity, erectile function, satisfaction
Musi et al. (2018) [20]	Author-developed questionnaire	Non-validated/Semi-quantitative	Erectile function, satisfaction, penile length, sensitivity, cosmetic perception
Skeppner et al. (2008) [21]	Structured interview including *LiSat-11*	Partially validated/Mixed	Sexual activity, satisfaction, life satisfaction
Skeppner et al. (2015) [22]	*IIEF-5*, *LiSat-11*, *HADS*	Validated/Quantitative	Erectile and sexual function, psychological well-being
Windahl et al. (2004) [23]	53-item structured interview including *IIEF-5* and *LiSat-11* items	Partially validated/Mixed	Erectile function, sexual satisfaction, ejaculation, dyspareunia, cosmetic perception
Kokorovic et al. (2021) [24]	Institutional questionnaire	Non-validated/Quantitative	Penile preservation, recurrence, functional status
Shaker et al. (2023) [25]	Patient interview	Non-validated/Qualitative	Erectile function, sexual activity

**Table 5 cancers-17-03737-t005:** Summary of individual study results. Synthesized data from individual studies are presented, covering functional outcomes (e.g., erectile function, sexual satisfaction), cosmetic outcomes, penile preservation, recurrence rates, and follow-up duration.

Study	Schlenker Study (2010) [17]	Tewari Study (2007) [18]	Frimberger Study (2002) [19]	Conejo-Mir Study (2005) [16]	Skeppner Study (2015) [22]	Skeppner Study (2008) [21]	Bandieramonte Study (2008) [15]	Musi Study (2018) [20]	Shaker Study (2023) [25]	Windahl Study (2004) [23]	Kokorovic Study (2021) [24]
Sample Size	54	32	29	8	29	46	224	26	20	67	58
Laser Type	Nd:YAG	Nd:YAG	Nd:YAG	CO_2_	Combined CO_2_, Nd:YAG	Combined CO_2_, Nd:YAG	CO_2_	Tm:YAG	Combined CO_2_, Nd:YAG	Combined CO_2_, Nd:YAG	KTP/CO_2_ or OSS + KTP
Treatment Protocol	(30–50 W power, 100 s, 3 mm margin), acetic-acid mapping, circumcision in all cases.	laser coagulation of tumor bed, circumcision in all cases	coagulation with 3 mm safety margin; acetic acid mapping before treatment.	Super-pulsed mode, 5–8 watts; 8–10 mm margin	N/A	N/A	Peniscopically controlled, with adjunctive vaporization of lesion margins	RevoLix 200 W continuous-wave laser, 360 μm fiber, 15–20 W power; safety margin of 3 mm vaporized	N/A	N/A	Laser monotherapy or combination used with acetic acid mapping
Functional Outcome	N/A	100% could urinate in standing posture	N/A	100% preserved urinary function.	47.6% dyspareunia before treatment, reduced to 12% at follow-up	N/A	Reported “excellent”	N/A	N/A	N/A	N/A
Scarring and Discoloration	Minimal	Minimal	Minimal	No hyperplastic scars	N/A	Minimal	Minimal	Minimal	N/A	No hyperplastic scars or gross deformities.	Minimal
Satisfaction with Cosmetic Outcome	80%	100%	100%	“High” satisfaction with cosmetic results “excellent cosmetic outcome” (physicians’ opinion)	N/A	Reported “High”	Reported “Excellent”;	N/A	N/A	78% satisfied/very satisfied; 100% retained a normal-appearing glans and meatus.	N/A
Tissue Preservation	N/A	100%	100%	100%	Penile preservation achieved in “most” cases	100%	96.0%; 4.0% required amputation.	100%	100%	100%	100%
Body Image Issues	“Superior” psychological outcomes compared to penectomy	100% satisfied	“Superior” psychological outcomes compared to penectomy		N/A	Comparable to general population	N/A	N/A	N/A	N/A	N/A
Penile Length Changes	“Superior” than in penectomy.	N/A	N/A	N/A	N/A	N/A	preserved penile form and curvature	78.2% no change	N/A	N/A	N/A
Outcomes Related to Sexual Satisfaction											
Sexual Satisfaction	59.3% of sexually active patients before surgery remained active post-treatment; “Superior” compared to amputation	75% reporting normal sexual satisfaction.	“Excellent” orgasmic function; 66.7% patients reported regular sexual activity	“High”	61% satisfied with sexual life before treatment, 32% after treatment (*p* = 0.039). 19% of patients stopped penetrative sex, but engaged in other forms of sexual activity Patient’s sexual partner satisfaction: low sexual desire before and after (8/29 then 9/29), decreased female lubrication (5/29 to 2/29), and partner sexual satisfaction remained the same.	65% of sexually active men before treatment resumed sexual activity Decreased manual stimulation/caressing and fellatio; 13.0% were sexually inactive; 21.7% did not resume sexual activities; 50.0% resumed intercourse; 95.7% did not change masturbation habits	“Satisfactory”	82.6% resumed intercourse; 13% did not continue intercourse 56.5% reported an impact; 43.5% reported no change	“Higher” in laser ablation and circumcision groups compared to glans excision	50% satisfied/very satisfied; 72% considered their sexual life to be as good as they wanted	N/A
Libido Changes	N/A	N/A	33% had no desire	N/A	Decreased libido in 45% before treatment, 34% after treatment	N/A	N/A	N/A	N/A	Unchanged in 80%, decreased in 17%, increased in 2.2%.	N/A
Time to Sexual Activity Resumption	N/A	N/A	N/A	N/A	Within one year 53% resumed intercourse; 19% resumed non penetrative sex.	6% within weeks; 59% within months.	N/A	60% had sexual intercourse within a month. 52.2% resumed erections within a week; 15.4% resumed erections after a month	N/A	75% had resumed sexual activity within the median 3-year follow-up	N/A
Erectile Function	59.3%	71.9% reported normal erectile function	33% reported erectile dysfunction	No erectile dysfunction reported.	IIEF-5 score ≥22 in 48.3% before, after: 34.5% at 1-year follow-up.	Reduced	No erectile dysfunction reported.	Unaltered post-treatment	N/A	72% unaltered; 22% decreased; 6% improved.	N/A
Penile Sensation	Self-reported sensitivity not or only slightly impaired	N/A	N/A	N/A	51.7% of patients reported decreased	N/A	Preserved in 100%	26.1% maintained; 56.5% improved; 17.4% worsened	N/A	N/A	N/A
Outcomes Related to Life Satisfaction											
Mental Health	Some patients experienced psychological distress due to recurrence	N/A	No suicidal thoughts reported	N/A	Factors of life satisfaction either improved slightly or remained unchanged. Anxiety in 17% of partners before treatment, 0% at follow-up	50.0% of patients satisfied with whole life, comparable to the general population	N/A	N/A	N/A	N/A	N/A
Psychological Counseling Received	N/A	N/A	None of the patients required	N/A	24.1% of patients and 17.2 partners discussed sexual issues	None. “Patients were somewhat less satisfied with their psychological health”	N/A	N/A	N/A	N/A	N/A
Post-Treatment Complications and Adverse Effects											
Post-Treatment Pain or Complication	N/A	No reported need for parenteral analgesics	No major complication.	Minimal; No major complications. Full re-epithelialization within 14–28 days	N/A	No major complications.	Minimal; pain resolved within 3 days post-op. No infections reported, minor local edema	Mild discomfort in external meatus cases. Minimal; mild edema of the prepuce and	No major complications or adverse effects reported	Dyspareunia reported in 10.4%.	N/A
Short vs. Long-Term Recovery	Fast recovery but required long-term follow-up due to late recurrences	Fast healing with epithelialization completed in 7–9 weeks	N/A	Full healing within 14–28 days	N/A	N/A	Healing completed in 6 weeks; no need for additional treatment in most cases	Healing by secondary intention in 5 weeks; no major complications	N/A	Full healing typically within 3 months	N/A
Adverse Effects and Complications	N/A	N/A	No major adverse effects reported.	Recurrence reported in the meatal area	Dyspareunia (11.8%)	No major adverse effects reported.	No major adverse effects reported.	Mild pain while urinating for a week	No major adverse effects reported.	N/A	N/A
Additional Treatments	Radical circumcision for additional safety and hygiene	None specified	Groin dissection in 10 of 12 patients (83%) with invasive tumors	None specified	None specified	Lymph node dissection in 16/46 patients (34.8%). Chemotherapy in 1/46 patients (2.2%). Radiotherapy in 1/46 patients (2.2%)	Reductive chemotherapy in select cases (exophytic tumors)	None specified	None specified	17/46 (37.0%) bilateral inguinal lymph node dissection; 1/46 (2.2%) received adjuvant chemotherapy	Inguinal lymphadenectomy in high-risk cases
Outcomes Related to Oncological Control											
Recurrence Rate	16/39 patients (42%); mean: 53 months	2/32 patients; 6.25%, occurring at 48–60 months	T1: 1 patient; CIS: 1 patient (2/29 patients; 6.9%)	12.5% (1/8 patients); at 1 year	N/A	8/46 patients (17.4%)	39/224 patients; (17.5%) at 10 years	17.4% (4/23 patients), including 13.0% invasive recurrence	recurrence: 4/20 patients (20.0%).	13/67 patients (19.4%); (10 successfully re-treated with laser	KTP/CO_2_—2/8 patients; 25.0% OSS + KTP—10/50 patients (20.0%)
Follow-Up Duration mean/median	87 months (range: 9–366)	70 months (range: 6–120)	46.7 months (range: 6–180)	120 months	12 months	54 months (range: 6–180)	66 months (range: 35–132)	24 months (range: 15–30)	57 months	36 months (range: 6–15)	28 months (range: 0.2–188)

**Table 6 cancers-17-03737-t006:** Summary of reporting biases. The degree of selective reporting across studies was evaluated, especially regarding sexual function, recurrence, and study methodology, categorized by laser type (CO_2_, Nd:YAG, Tm:YAG, and others).

Laser Therapy	Sexual Function Reporting Bias	Recurrence Reporting Bias	Other Bias Concerns	Overall Risk of Reporting Bias
CO_2_ Laser	No quantitative sexual function data	Inconsistent recurrence tracking	Short follow-up in some studies	Moderate to High
Nd:YAG Laser	No standardized assessments	Large variation in recurrence rates	High variability in follow-ups	High
Tm:YAG Laser	Only one study, qualitative function data	Short follow-up, underestimation likely	No control group	Very High
Miscellaneous	Sexual function data inconsistently reported	Wide variation in follow-up (12 months–15 years)	Subjective interviews	Moderate to High

**Table 7 cancers-17-03737-t007:** Summary of certainty of evidence for sexual function outcomes. GRADE-based evaluation of the quality of evidence for key outcomes such as erectile function, libido, orgasm, penile preservation, recurrence rates, and patient satisfaction is provided.

Outcome	Certainty of Evidence	Justification
Erectile function preservation	Low	No standardized assessments, inconsistent self-reported data
Libido changes	Very Low	Only qualitatively reported in select studies, no objective measures
Orgasmic function	Very Low	No direct measurement in any study
Penile preservation success	Moderate	Consistently reported across studies (96–100%)
Time to resumption of sexual activity	Very Low	Reported inconsistently, no standardized measurement
Short-term recurrence (<2 years)	Moderate to Low	Reasonably well reported but varies across studies
Long-term recurrence (>5 years)	Low	Follow-up durations are inconsistent; late recurrences may be underreported
Overall recurrence-free survival	Low	Large variation in recurrence estimates across laser therapies
5-year overall survival	Moderate	Reported in some but limited data
5-year disease-free survival	Low	Inconsistently reported across studies
10-year survival	Very Low	No study systematically tracked survival beyond 5 years
Overall patient satisfaction	Low	Reported qualitatively but inconsistently
Quality of life post-treatment	Very Low	Lack of structured QoL assessments
Partner satisfaction	Very Low	No data available in most studies
Sexual function	Low	No standardized assessments, qualitative data mostly
Recurrence rates	Low	High variability, inconsistent follow-up durations
Oncologic control	Low to Moderate	Few survival data points, no long-term tracking

## Data Availability

All data used in this review were extracted from published studies and are presented in the text, tables, and Appendix A. Extraction sheets and analytic code are available from the corresponding author upon reasonable request.

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
