# Peer review of "Sexual Function After Laser Therapy for Penile Cancer: A Systematic Review"

_cancers, 2025, doi:10.3390/cancers17233737_

Round 1
Reviewer 1 Report
Comments and Suggestions for Authors
Please cite main articles in the filed such as: Fallara, Giuseppe et al. “Oncological and Functional Outcomes of Penile Shaft Sparing Surgery for Localised Penile Cancer: A Systematic Review.” European urology focus vol. 11,1 (2025): 46-61. doi:10.1016/j.euf.2024.05.004
The work lack PROSPERO registration which is foundamental for quality check assessment in systematic review.
What is the idea bheind the study? Why there should be any difference in sex function because of the use of different laser?
The same for Tissue Preservation chapter. What do you mean? It is not the laser type, but the surgical indication based on the cT stage that determines how much tissue you can preserve.
In addition, given the title of you article, i.e. Sexaul Fuinction etc. why do you stress so much the oncological outcomes? I think you can redice the length of the paper and elimante all the irrilevant paragraphs.
Author Response
Comment 1:
“Please cite main articles in the field such as: Fallara, Giuseppe et al. “Oncological and Functional Outcomes of Penile Shaft Sparing Surgery for Localised Penile Cancer: A Systematic Review.” European Urology Focus vol. 11,1 (2025): 46-61. doi:10.1016/j.euf.2024.05.004.”
Response 1:
Thank you for this valuable suggestion. We agree that the study of Fallara et al. and other recent systematic reviews and guidelines represents a major contribution to the contemporary literature on penile-sparing surgery. To acknowledge this, we have now cited Fallara et al. in the Introduction to emphasize the growing body of evidence supporting organ-preserving approaches in penile cancer.
We would like to clarify that our systematic review aims to complement their findings by addressing a specific gap in the literature. While previous reviews have broadly examined penile-sparing techniques, our work focuses exclusively on laser-based modalities and their effects on sexual and functional outcomes, thereby expanding upon prior findings within a narrower and more clinically focused scope.
This addition has been incorporated in Section 1 (Introduction), page 2, paragraph 3, lines 68-73. Furthermore, the relevance of recent systematic reviews and guidelines is also acknowledged in the Materials and Methods section. We appreciate your comment, which has helped us strengthen the connection between our focused analysis and the broader body of evidence in this field.
Text in the article:
- Section 1 (Introduction), page 2, paragraph 3, lines 68-73: This paradigm shift has led to the increasing adoption of penile-sparing surgery, balancing effective tumor eradication with improved postoperative quality of life, including sexual and urinary outcomes. Recent systematic reviews, such as that by Fallara et al. (2025), have reinforced the expanding role of penile-sparing approaches in localized disease, demonstrating favorable oncologic outcomes across more than 10,000 patients. [5,6]. Despite this increasing interest in organ-sparing strategies for penile cancer, the current literature lacks a cohesive synthesis of how laser-based treatments impact long-term sexual function and patient-reported outcomes. Many existing studies prioritize recurrence and survival metrics while underreporting functional and psychosocial endpoints, such as erectile function, orgasm, and patient satisfaction.
- Section 2.1 - Materials and Methods, Page 3, Lines 124-127: For the purpose of this review, while systematic reviews and guidelines offered in-valuable insight, they were excluded due to lack of primary data. Instead, their references were manually screened for the primary sources of their data.
Comment 2:
“The work lacks PROSPERO registration, which is fundamental for quality check assessment in systematic reviews.”
Response 2:
We thank the reviewer for this valuable observation. We agree with the reviewer. Therefore, we have taken steps to further ensure methodological transparency and reproducibility. Although this review does not meet the inclusion criteria for PROSPERO registration, we have now retrospectively registered the study protocol on the Open Science Framework (OSF). The OSF record provides a publicly accessible summary of the study objectives, inclusion criteria, and planned analyses, thereby fulfilling the principles of open and transparent research. The registration details have been added to the Methods section. This addition can be found in Section 5 (Other Information), page 21, paragraph 1, lines 578-583. This update strengthens the methodological quality of our review and aligns it with current best practices in reporting and for that we once again thank the reviewer.
Text in the article:
- Section 5 (Other Information), page 21, paragraph 1, lines 578-583:
“The study protocol was retrospectively registered on the Open Science Framework (OSF), under a CC BY 4.0 license to enhance reproducibility and public accessibility (https://doi.org/10.17605/OSF.IO/GJAZ6). OSF registration was selected as an appropriate alternative to PROSPERO, as the review focuses on functional and quality of life outcomes rather than interventional therapies, which fall outside PROSPERO’s eligibility criteria.”
Comment 3:
“What is the idea behind the study? Why should there be any difference in sexual function because of the use of different lasers? Do their tissue-interaction profiles have clinically meaningful implications?”
Response 3:
Thank you for pointing this out. We agree with this comment and we have worked to minimize the ambiguity in the rationale and to further link previous findings with our research. We have now clarified the scientific rationale for comparing different laser modalities and expanded the Discussion to explain whether their tissue-interaction profiles have clinically meaningful implications. We have also included studies regarding the mechanisms of action of each of the lasers to bring the topics closer.
A new paragraph has been added to the Introduction (page 2, paragraph 5, lines 80-96) and another explanatory paragraph has been inserted in the Section 4 - Discussion (page 20, paragraph 2, lines 516-531). These additions describe how each laser interacts with tissue, how each laser has different physical properties, and how these physical properties may influence biological tissue and hence the postoperative outcomes.
Text in the article:
- Introduction (page 2, paragraph 5, lines 80-96):
“Additionally, there is no established consensus on which laser modalities best balance oncologic control with quality-of-life preservation, especially in early-stage disease. Each laser modality interacts with biological tissue through distinct photothermal mechanisms, which may influence postoperative healing and functional outcomes. The COâ‚‚ laser emits infrared light at 10.6 μm, resulting in superficial vaporization with a penetration depth of approximately 0.05 mm, with a thermal coagulation zone of ap-proximately 0.5 mm and minimal collateral thermal injury, allowing precise excision and rapid mucosal recovery [7]. This produces an extremely narrow zone of thermal damage, but the residual thermal energy is not enough to provide hemostasis in vascularized tissues. In contrast, the Nd: YAG laser belongs to the group of solid-state lasers which operates at 1.064 μm with deeper penetration (4-6 mm) and greater coagulative necrosis, which, while advantageous for hemostasis and oncologic control, may increase the risk of damage to adjacent neurovascular structures relevant to erectile and sensory function [8,9]. The Tm: YAG laser (1.94 μm) exhibits intermediate absorption and shallow penetration (~0.3-0.5 mm), combining efficient cutting with excellent coagulation and limited thermal spread [9,10]. These biophysical differences may translate into variable effects on scarring, sensitivity, and erectile function after treatment. This gap complicates clinical decision-making, leaving patients with limited evidence to guide treatment decisions.”
- Section 4 - Discussion (page 20, paragraph 2, lines 516-531):
“The comparative findings across laser modalities suggest that their distinct tis-sue-interaction profiles may have modest but clinically plausible implications for post-operative functional outcomes. The COâ‚‚ laser, characterized by shallow penetration (≈0.05 mm) and a narrow thermal coagulation zone, achieved the highest rates of erectile function preservation and the most rapid recovery of sexual activity [7,15,16]. In contrast, the Nd: YAG laser, with deeper optical penetration (4-6 mm) and greater coagulative necrosis, demonstrated slightly lower sexual satisfaction rates (59-72%)[17-19], findings consistent with its broader thermal spread and potential to affect subepithelial neuro-vascular structures [8,9]. The Tm :YAG laser, with intermediate absorption depth (0.3-0.5 mm), produced outcomes between those of COâ‚‚ and Nd :YAG with 82% of patients re-turning to sexual intercourse, aligning with its balanced cutting-coagulation properties [9,20]. Although these differences correspond to known biophysical mechanisms, current evidence remains suggestive rather than definitive, given the heterogeneity of study designs and the absence of standardized sexual-function assessment tools. Larger comparative studies are needed to determine whether these optical distinctions translate into durable clinical advantages."
Comment 4:
“The same for the Tissue Preservation chapter. What do you mean? It is not the laser type, but the surgical indication based on the cT stage that determines how much tissue you can preserve.”
Response 4:
We thank the reviewer for this important clarification. We fully agree that the degree of penile tissue preservation is determined primarily by the tumor’s clinical stage (cT) and the corresponding surgical indication, rather than by the laser modality itself. To reflect this, we have revised Section 3.3.5 (now titled “Extent of Tissue Preservation”) to explicitly state that oncologic indication dictates the extent of resection, while laser modalities influence precision, margin control, and the extent of collateral thermal injury. We also clarified that the high preservation rates (>96%)observed across laser-treated cohorts were anticipated due to the predominance of CIS, Tis, and T1 tumors, yet the near-complete preservation supports the notion that laser therapy causes no appreciable additional tissue loss beyond what is required for oncologic clearance. This revision clarifies the intended meaning and aligns the section with accepted clinical reasoning. The changes can be found in Section 3.3.5 (Extent of Tissue Preservation), page 15, lines 338-353.
Additionally, we took a step further and eliminated this ambiguity anywhere else within the manuscript. Throughout, we now replaced “tissue preservation” to “extent of tissue preserved” or “percentage of penile tissue undamaged”. We sincerely thank the reviewer for this comment.
Text in the article:
- Section 3.3.5 (Extent of Tissue Preservation), page 15, lines 338-353:
“While the extent of penile tissue preservation is primarily determined by the tumor’s clinical stage (cT) and corresponding surgical indication, the uniformly high preservation rates observed across laser-treated cohorts (>96%) suggest that laser techniques allow for precise ablation with minimal collateral thermal injury. In COâ‚‚ laser studies, organ preservation ranged from 96% to 100% [15,16], while Nd:YAG and Tm:YAG series consistently reported 100% preservation in appropriately selected early-stage patients [17–19]. Comparable results were also achieved with combined COâ‚‚ + Nd:YAG therapy [21–23]. While such high preservation rates were anticipated given that most included patients had carcinoma in situ (CIS), Tis, or T1 disease, the near-complete preservation observed suggests that the lasers caused no appreciable additional thermal damage beyond the tissue margins required for oncologic clearance. Collectively, these findings indicate that, although tumor stage dictates the extent of resection, the laser’s precision and limited depth of thermal injury contribute to excellent immediate tissue preservation and favorable cosmetic outcomes in early-stage penile cancer. Detailed data on penile preservation outcomes are presented in Supplementary Table S7.”
Comment 5:
“In addition, given the title of your article, i.e., Sexual Function…, why do you stress so much the oncological outcomes? I think you can reduce the length of the paper and eliminate all the irrelevant paragraphs.”
Response 5:
We thank the reviewer for this comment. We agree that the primary focus of the manuscript should remain on sexual and functional outcomes, consistent with the article’s title. However, we also recognize that oncologic control is a fundamental prerequisite for evaluating the safety and durability of any penile-sparing technique. To address this point and streamline the text, we have condensed and reorganized the sections describing recurrence (previously 3.3.9), oncologic control (previously 3.3.8), and survival outcomes into a single subsection now titled “3.3.6. Clinical Outcomes.” This revision eliminates redundancy and improves readability while retaining essential data needed to contextualize the functional findings.
At the same time, we would like to note that another reviewer requested a more detailed presentation of oncologic data to ensure balanced interpretation. We therefore reached a compromise that maintains a concise yet comprehensive overview of the relevant evidence. The revised section integrates recurrence, survival, and long-term control data into one coherent paragraph, emphasizing their collective importance for safe tissue preservation and postoperative sexual recovery. The updated text can be found in Section 3.3.6 (Clinical Outcomes), page 15, lines 354-373.
Text in the article:
- Section 3.3.6 (Clinical Outcomes), page 15, lines 354-373:
“Oncologic control following laser-based penile-sparing therapy was favorable across all modalities. Recurrence rates varied across laser types, influenced by differences in tumor staging and follow-up duration. CO2 laser studies reported relatively consistent outcomes: 12.5% in Conejo-Mir et al. and 17.4% in Bandieramonte et al. [15,16]. Nd:YAG studies showed greater variability, with one reporting a high recurrence rate of 42.6%, likely influenced by longer follow-up and inclusion of higher-stage tumors [17], while others [18,19] reported substantially lower rates of 6.25% and 6.9%, respectively. The Tm:YAG study reported a recurrence rate of 17.4%, including 13.0% invasive recurrences, comparable to CO2 laser outcomes but based on a shorter follow-up period [20]. Five-year survival outcomes following laser-based therapies were favorable across modalities. Five-year survival was reported 100% in one study for CO2 laser therapy [16], Across studies employing Nd:YAG laser therapy for early-stage penile carcinoma, 5-year diseasespecific survival ranged from 98% to 100%. [17–19], and 95.6% for combined CO2 + Nd:YAG therapy, with disease-specific survival ≥95% where reported [21–23]. Tm:YAG therapy achieved 100% survival in early-stage disease at 2 years [20]. Overall, all laser types demonstrate acceptable oncologic control in early-stage disease but highlight the importance of rigorous, long-term follow-up to detect delayed recurrences and validate long-term efficacy. Details regarding the recurrence and the clinical outcomes are given in Supplementary Table S8.”
Additional comments for the reviewer:
Lastly, In response to the reviewer’s ratings and comments , we have revised all tables to enhance clarity, readability, and consistency throughout the manuscript. Redundant or overlapping information has been removed to reduce clutter, and formatting has been standardized to ensure uniform presentation of study characteristics, outcomes, and methodological details. Headings, abbreviations, and column structures have been harmonized for easier comparison across tables. Despite our best efforts, in some instances the primary data were not reported uniformly across studies. In the interest of transparency, we have chosen to clearly display these variations in reporting rather than omit or modify them. These collective improvements ensure that the tables present the data in a concise, coherent, and visually accessible manner while preserving the integrity and completeness of the extracted evidence.
Reviewer 2 Report
Comments and Suggestions for Authors
The aim of this paper was to systematically review sexual function after laser treatment for penile cancer by exploring data from previously conducted studies. The introduction provided sufficient information about penile cancer and laser treatment for early stage disease. All provided data clearly explained the rationale for conducting this review exploring generally less covered field in reporting sexual outcomes after organ-spearing treatment.
Materials and Methods meticulously explained the process of selecting appropriate previously published papers. Eligibility criteria were adequate, and outcome measures, search strategy, selection process, data collection process and bias and quality assessment risk were described in detail.
The results were presented in a logical and sound manner with clear explanation. Maybe more attention could be brought to the Table 1. and uniformity of reported data. For example when reporting sample size of study by Frimberger there is additional stratification of subjects by cancer stage that is already reported in cancer stage column. Also age range and mean age was not reported in uniform manner. Authors paid special attention to the risk of bias in selected studies and appropriately used Newcastle-Ottawa Scale to assess it. Certainty of evidence was also emphasized.
Discussion was presented in sound manner pointing out great deal of limitations in comparing the heterogeneous data and lack of consistent reporting in selected papers. Majority of previously conducted studies did not use patient reported outcomes and validated questionnaires to measure sexual function, satisfaction and overall quality of life, so the authors adequately recognized need for prospective or randomized trials to further assess sexual function after organ-spearing procedures for penile cancer.
The manuscript is clear, relevant and exploring a topic that follows modern views that not oncologic outcome is important, but also patient reported satisfaction. It is presented in a well-structured manner and is scientifically sound, providing advancement of current knowledge. Figures and tables properly show the data. Data is consistently interpreted throughout the manuscript. Conclusions are supported by the results and data provided. Citations used are appropriate.
Author Response
Comment 1:
“Maybe more attention could be brought to the Table 1. and uniformity of reported data. For example when reporting sample size of study by Frimberger there is additional stratification of subjects by cancer stage that is already reported in cancer stage column. Also age range and mean age was not reported in uniform manner.”
Response 1:
Thank you for your heartfelt review and for pointing this out. We share your opinion regarding the earlier presentation of our tables. We have therefore worked to ensure the formats of our tables remain consistent throughout. Despite our best efforts, in some instances the primary data is not recorded uniformly. In the interest of transparency, we decided to clearly display these variations in reporting. Having said that, we made our best effort to ensure the tables are easy to follow, reduce their clutter, and increase uniformity. The changes are now reflected in the mentioned Table 2 on page 7. Finally, we took a step further and held all other tables to the same standard and rigor. The remaining tables have all been updated using the same criteria. We sincerely thank you for your attention to details and helping us improve our presentation of data.
Reviewer 3 Report
Comments and Suggestions for Authors
This systematic review addresses an important and clinically relevant topic regarding sexual function outcomes following laser therapy for penile cancer. The work follows appropriate PRISMA 2020 guidelines and attempts to synthesize evidence across multiple laser modalities (CO2, Nd:YAG, Tm:YAG). However, the manuscript requires substantial revisions to improve methodological rigor, clarity, and alignment with current evidence standards before acceptance.
1.The authors acknowledge that "sexual outcomes were inconsistently reported, often using non-validated tools." This is a critical methodological weakness that undermines the validity of conclusions. Thus, authors should provide a detailed table in the revised manuscript categorizing studies by outcome measurement tools used
2.Clearly distinguish between Validated instruments (e.g., International Index of Erectile Function - IIEF);Non-validated or author-derived instruments;Qualitative assessments;
3.While the review focuses on sexual function, the balance between functional preservation and oncological control is insufficient.
4.The authors employ GRADE methodology but provide limited detail on certainty of evidence ratings for individual outcomes.
6.While the Newcastle-Ottawa Scale (NOS) was used, the manuscript lacks transparent reporting of individual study ratings.
7.The search was limited to English-language publications, potentially excluding relevant non-English literature from major urological centers (e.g., European centers with non-English publications)ï¼› The combination of MeSH terms and free-text terms should be explicitly detailed for each database in the main manuscript (currently referenced only to supplementary material)ï¼›No mention of searching conference proceedings (ASCO, EAU, AUA) for recent abstracts
Author Response
Comment 1:
“The authors acknowledge that "sexual outcomes were inconsistently reported, often using non-validated tools." This is a critical methodological weakness that undermines the validity of conclusions. Thus, authors should provide a detailed table in the revised manuscript categorizing studies by outcome measurement tools used.”
Response 1: We would like to sincerely thank the reviewer for their constructive and kind review. We fully agree with the reviewer that the heterogeneity of outcome assessment represents a key methodological limitation that must be transparently reported. To address this, we have added a new table summarizing the type of instrument used in each included study, categorized as validated (e.g., IIEF, LiSat-11, HADS), non-validated/author-derived, or qualitative assessments. This table clarifies the extent of methodological variation in sexual function measurement and supports the discussion of evidence certainty. The new material has been added as Table 4 (page 9) and referenced in Section 3.3.1. (Heterogeneity), Page 9, Paragraph 2, lines 280-281.
Text in the article:
- Section 3.3.1. (Heterogeneity), Page 9, Paragraph 2, lines 280-281:
“A detailed summary of the assessment instruments used in each study is presented in Table 4.”
- Table 4:
Outcome measurement instruments used across included studies for assessment of sexual and functional parameters.
Comment 2:
“Clearly distinguish between validated instruments (e.g., International Index of Erectile Function - IIEF), non-validated or author-derived instruments, and qualitative assessments.”
Response 2:
As a continuation of the previous comment we thank the reviewer and agree that the heterogeneity of outcome assessment can be reported more thoroughly. To address this, we have added both a dedicated explanatory paragraph summarizing the instruments used to evaluate sexual and functional outcomes across all included studies and the previously mentioned table 4. The revisions appear in Section 3.3.1. (Heterogeneity), Page 9, Paragraph 2, lines 271-280. We have further addressed this by defining validated and non-validated instruments in Section 2.6. (Data Items), Page 5, Paragraph 2, lines 171-183.
The new paragraph highlights when studies employed validated questionnaires (e.g., International Index of Erectile Function, IIEF), and when non-validated, author-derived tools or qualitative patient interviews was used. This clarification emphasizes the variability in measurement quality and its implications for evidence certainty.
As mentioned earlier, Table 4 categorizes each study according to the type of assessment tool used (validated, non-validated, or qualitative) and the specific functional domains evaluated.
Text in the article:
- Section 2.6. (Data Items), Page 5, Paragraph 2, lines 171-183:
“Data regarding study design, patient characteristics, tumor stage, laser modality, and outcome measures were systematically extracted. Sexual and functional outcomes were categorized according to the assessment instruments used to enhance comparability across studies. These included: (1) validated tools, such as the International Index of Erectile Function (IIEF), Life Satisfaction Checklist-11 (LiSat-11), Hospital Anxiety and Depression Scale (HADS) or the European Organization for Research and Treatment of Cancer Quality of Life Questionnaire-Core 30 (EORTC QLQ-C30) if available; (2) non-validated or author-derived instruments, including study-specific questionnaires and numerical rating scales without prior validation; and (3) qualitative assessments, encompassing structured or semi-structured interviews and patient self-reports describing sexual activity, satisfaction, or changes in erectile performance. This classification allowed consistent evaluation of methodological heterogeneity and facilitated interpretation of sexual function outcomes across different studies. “
- Section 3.3.1. (Heterogeneity), Page 9, Paragraph 2, lines 271-280:
“The outcome instruments used across studies were highly heterogeneous, reflecting substantial methodological variability in how sexual function was defined and assessed. Only a minority of studies employed validated questionnaires, such as the IIEF, LiSat-11, or HADS, to quantify erectile or sexual outcomes objectively. The majority relied on non-validated, author-developed questionnaires or qualitative patient interviews, often lacking standardized scoring systems or reference thresholds. In several retrospective series, sexual function was inferred from patient self-reports or non-specific terms, which limits comparability and interpretability across studies. These methodological inconsistencies reduce the certainty of pooled evidence and underscore the need for standardized, validated outcome measures in future research.”
Comment 3:
“While the review focuses on sexual function, the balance between functional preservation and oncological control is insufficient.”
Response 3:
We thank the reviewer for this observation. We agree on the importance of balanced oncologic control and their interplay with functional preservation. Moreover, we agree with the reviewer that sexual function preservation cannot happen at the cost of oncological control. Accordingly, we have expanded the Discussion to include a focused synthesis of recurrence, survival, and management outcomes across different laser modalities, while clearly contextualising these within the framework of functional preservation. At the same time, we have taken into account another reviewer’s recommendation to maintain conciseness, given the primary focus of this review on sexual and functional outcomes, and to reduce the length of discussion regarding the oncological outcomes. We therefore had to reach a compromise. The revised section therefore integrates the oncologic dimension in a concise and balanced manner, emphasising that the long-term success of laser therapy depends on meticulous patient selection, structured surveillance, and multidisciplinary care. The updated text appears in Section 4 (Discussion), page 19, paragraph 4-6, lines 484-502.
Text in the article:
- Section 4 (Discussion), page 19, paragraph 4-6, lines 484-502:
“From an oncological perspective, laser-based penile-sparing therapy demonstrates recurrence and survival outcomes that are broadly comparable to other conservative and radical approaches when applied in appropriately selected patients. Across the included studies, local recurrence rates ranged from 6-19% [5]. Most recurrences were superficial and successfully managed with repeat laser ablation, thereby preserving penile anatomy and avoiding the morbidity of salvage partial or total penectomy. The reported five-year overall survival exceeded 90 % across. Although recurrence rates may be somewhat higher than those seen with more radical procedures, the availability of repeat laser treatment and preservation of tissue integrity allow for flexible, staged management.
However, these favourable results are contingent upon rigorous and sustained postoperative surveillance, which is essential for detecting and managing local recurrence at an early stage. Limited access to specialist follow-up care, resource constraints, or patient non-adherence to long-term monitoring may undermine these outcomes. Moreover, patients who prioritise definitive reassurance over functional cost may opt for more radical options.
Collectively, the evidence suggests that laser therapy achieves a favourable balance between oncologic control and functional preservation in early-stage penile cancer, but its long-term success depends on meticulous patient selection, adherence to structured follow-up, and shared decision-making within a multidisciplinary care framework. [29].”
Comment 4:
“The authors employ GRADE methodology but provide limited detail on certainty of evidence ratings for individual outcomes.”
Response 4:
We thank the reviewer for this important comment. We agree that in addition to our GRADE Summary table (Table 7 - GRADE-based evaluation of the quality of evidence), some additional methodological detail would enhance the clarity of these ratings. Accordingly, we have expanded Section 3.5 (Certainty of Evidence) to describe the specific criteria used for downgrading and upgrading evidence across the five GRADE domains (risk of bias, inconsistency, indirectness, imprecision, and publication bias) and to indicate that observational studies were initially rated as low certainty. We also specified that upgrades were applied when large effect sizes or consistency across studies were observed. In addition, Table 7 is kept to maintain transparency and interpretability of the certainty assessment and to provide more details on our GRADE scoring. These changes can be found in Section 3.5 (Certainty of Evidence), page 17, paragraph 1, lines 425-433 and Table 7 (page 17).
Text in the article:
- Section 3.5 (Certainty of Evidence), page 17, paragraph 1, lines 425-433:
“Certainty of evidence for each primary and secondary outcome was assessed using the Grading of Recommendations, Assessment, Development, and Evaluation (GRADE) framework in accordance with Cochrane and GRADE Working Group guidance. Evidence derived from observational studies was initially rated as low certainty and subsequently downgraded or upgraded based on five domains: risk of bias, inconsistency, indirectness, imprecision, and publication bias. Upgrading was considered when consistent large effects or dose-response relationships were observed across multiple studies. Final certainty were determined for each outcome, including erectile function, sexual satisfaction, “penile preservation, recurrence, and survival.
- Table 7 (page 17): Summary of certainty of evidence for sexual function outcomes. GRADE-based evaluation of the quality of evidence for key outcomes such as erectile function, libido, orgasm, penile preservation, recurrence rates, and patient satisfaction is provided.
Comment 5 (numbered 6):
While the Newcastle-Ottawa Scale (NOS) was used, the manuscript lacks transparent reporting of individual study ratings.
Response 5:
We thank the reviewer for this comment. We would like to clarify that a full Newcastle-Ottawa Scale (NOS) assessment for each study was already included in the manuscript as Table 3, providing detailed domain-level ratings (selection, comparability, outcome), total scores, overall bias interpretation, and reviewer comments. To improve visibility, we have now explicitly referenced this table in both the Results (Section 3.2) and summarized the overall range of NOS scores in the text. We have additionally further added an explanation of how this NOS was calculated in the body of our manuscript. These changes can be found in Section 3.2 (Characteristic of Studies - Risk of Bias in Studies), page 8, paragraph 1, lines 256-257, and Table 3 (page 8). These additions enhance transparency and ensure the reader can readily interpret the methodological quality of each included study.
Text in the article:
- Section 2.7 – Methods (Risk of Bias and Quality Assessment), page 5, paragraph 2, lines 185-189:
“The risk of bias was assessed using the Newcastle-Ottawa Scale (NOS) for observational studies, covering three domains: selection (4 points), comparability (2 points), and outcome (3 points). Two independent reviewers conducted these assessments and resolved disagreements through discussion or consultation with a third reviewer.”
- Section 3.2 (Characteristic of Studies - Risk of Bias in Studies), page 8, paragraph 1, lines 256-257:
“A detailed breakdown of NOS scoring is presented in the risk of bias assessment in Table 3.”
- Table 3 (Page 8):
“Risk of bias assessment. The methodological quality of the included studies using tools such as the Newcastle-Ottawa Scale (NOS) is summarized, with commentary on potential biases and limitations.”
Comment 6 (numbered 7):
“The search was limited to English-language publications, potentially excluding relevant non-English literature from major urological centers (e.g., European centers with non-English publications). The combination of MeSH terms and free-text terms should be explicitly detailed for each database in the main manuscript (currently referenced only to supplementary material). No mention of searching conference proceedings (ASCO, EAU, AUA) for recent abstracts.”
Author Response 6:
We thank the reviewer for this observation and agree that additional clarification regarding our search strategy improves transparency and reproducibility.
The restriction to English-language publications was initially applied to ensure consistency in data extraction and quality appraisal. Nonetheless, to minimize potential selection bias, we have now revisited those articles that were earlier deemed ineligible due to linguistic issues. Therefore, we have screened non-English titles and abstracts using automated translation tools (Google Translate and Apple’s Safari’s Translate Extension) to identify potentially eligible records. We were prepared to involve a professional translator for full-text evaluation if necessary; however, this was not required, as no non-English studies met the inclusion criteria.
We have also expanded the description of our search methodology in the Materials and Methods section. The revised text now explicitly details the use of both Medical Subject Headings (MeSH) and free-text terms for each search concept (“penile cancer,” “laser therapy,” and “sexual function”), provides an illustrative PubMed search string, and clarifies that equivalent syntax was adapted for Embase, Scopus, and ScienceDirect. The full search codes are now available in Table 1 (Page 4).
Finally, we added that conference proceedings from EAU, AUA, and ASCO were screened for relevant recent abstracts not yet published in full text. These additions have been reflected in the PRISMA 2020 flow diagram and incorporated into the main manuscript in Section 2.1 (Eligibility Criteria), page 3, paragraph 1, Lines 125-127, and Section 2.3 (Search Strategy), page 4, paragraph 1, Lines 144-145. We once again thank the reviewer for improving the breadth of our study.
Text in the article:
- Section 2.1 (Eligibility Criteria), page 3, paragraph 1, Lines 125-127:
“Furthermore, articles written in languages other than English were initially translated using automated tools. If the preliminary assessment indicated potential, they would be sent for professional translation.”
- Section 2.3 (Search Strategy), page 4, paragraph 1, Lines 144-145:
“A systematic search was performed in accordance with the PRISMA 2020 guidelines. We searched PubMed, Embase, Scopus, Google Scholar, ClinicalTrials.gov, and ScienceDirect using a combination of Medical Subject Headings (MeSH) and free-text terms pertaining to penile cancer, laser therapy, and sexual function. A further search was conducted on conference databases such ASCO, EAU, and AUA for abstracts.”
- Table 1 (page 4): Detailed search strings and Boolean combinations employed in each database are presented.
Additional comments for the reviewer:
Lastly, In response to the reviewer’s ratings and comments , we have revised all tables to enhance clarity, readability, and consistency throughout the manuscript. Redundant or overlapping information has been removed to reduce clutter, and formatting has been standardized to ensure uniform presentation of study characteristics, outcomes, and methodological details. Headings, abbreviations, and column structures have been harmonized for easier comparison across tables. Despite our best efforts, in some instances the primary data were not reported uniformly across studies. In the interest of transparency, we have chosen to clearly display these variations in reporting rather than omit or modify them. These collective improvements ensure that the tables present the data in a concise, coherent, and visually accessible manner while preserving the integrity and completeness of the extracted evidence.
Reviewer 4 Report
Comments and Suggestions for Authors
Please address following comments.
- Results, 3.2.: Please describe types of studies (e.g., RCT, prospective study, retrospective study…)
- Discussion, the 1st paragraph: What is a basis of “excellent” preservation of…?
- Conclusions, the 1st sentence: What is laser therapy superior to? Please describe the comparator.
Author Response
Comment 1:
“Results, 3.2: Please describe types of studies (e.g., RCT, prospective study, retrospective study…).”
Response 1: Thank you for pointing this out. We agree with this comment. Therefore, we have added a summary describing the study designs represented in our review to clarify the methodological composition of the included studies. This paragraph has been also extended to bring attention to Table 2. Which included a detailed breakdown of each study’s characteristics, design, features, and main findings. This addition can be found in Section 3.2 (Study Characteristics), page 6, paragraph 2, lines 219-226.
Text in the article:
- Section 3.2 (Study Characteristics), page 6, paragraph 2, lines 219-226:
“Among the included studies, all except one were retrospective in design, reflecting the rarity of penile cancer and the predominance of single-center experiences. Specifically, eight studies were retrospective (including case series, cohort, and interview-based analyses), one was a prospective observational study, one was a retrospective cohort study, and one was a retrospective review. No randomized controlled trials (RCTs) were identified. This distribution underscores the descriptive and exploratory nature of the current evidence base for laser-based penile-sparing therapy. The study characteristics, study design, and main findings are summarized in Table 2.”
- Table 2 (page 7):
“Study characteristics and main findings. Details such as study design, sample size, laser type used, patient age, cancer stage, and key clinical outcomes across the included studies are included.”
Comment 2:
“Discussion, the 1st paragraph: What is a basis of “excellent” preservation of…?”
Response 2:
Thank you for this valuable comment. We agree that the statement required clarification with quantitative evidence. Therefore, we have revised the opening paragraph of the Discussion to specify the numerical basis for functional and anatomical preservation outcomes rather than using subjective terminology. The updated text now provides concrete numerical data when subjective adjectives are used. These changes can be found in Section 4 (Discussion), page 19, paragraph 1, lines 460-468.
In addition to this, we screened the text once more to ensure subjective terminology is either avoided or followed by statistical or numerical data. Where the subjective phrasing was employed by the primary sources of data, i.e. in studies, we outlined this using “quotation marks”.
Text in the article:
- Section 4 (Discussion), page 19, paragraph 1, lines 460-468:
“This review supports laser therapy as an effective penile-sparing option for patients with early-stage penile cancer, demonstrating high rates of functional preservation. Across included studies, erectile and sexual function were maintained in approximately 60-100% of patients, and penile tissue was successfully preserved or undamaged in 96-100% of cases, with most reports also describing favourable cosmetic results and minimal morbidity. These outcomes indicate that laser therapy provides reliable functional preservation while maintaining oncologic safety, aligning with the growing clinical emphasis on quality of life in penile cancer management.”
Comment 3:
Conclusions, the 1st sentence: What is laser therapy superior to? Please describe the comparator.
Response 3:
As an extension of the previous comment, we once again thank the reviewer for this observation. We agree that the comparator should be explicitly stated. The revised text now specifies that laser therapy offers superior functional outcomes and lower morbidity compared with conventional non-penile-sparing surgical procedures. This clarification has been added to the first sentence of the Conclusions section. The updated version can be found in Section 4.2 (Conclusions), page 21, lines 566-569.
Text in the article:
- Section 4.2 (Conclusions), page 21, lines 566-569:
“Laser therapy represents a functionally superior, low-morbidity alternative to conventional non-penile-sparing surgical approaches, including partial and total penectomy, for the management of early-stage penile cancer.”
Round 2
Reviewer 3 Report
Comments and Suggestions for Authors
Authors have solved all concerns.